# DNCs require more planning steps

## Abstract

A Differentiable Neural Computer (DNC) is a memory-augmented neural computation model capable of learning to solve complex algorithmic tasks, from simple sorting algorithms, through graph problems, to text question answering. In previous works, it was always given a small constant number of planning steps to complete its task. In this work, we argue that the number of planning steps the model is allowed to take, which we call "planning budget", is a constraint that can cause the model to generalize poorly and hurt its ability to fully utilize its external memory. By introducing an adaptive planning budget that scales with input size during training, the model is better able to utilize its memory space, and achieves substantially better accuracy on input sizes not seen during training. We experiment with Graph Shortest Path search, which has been used as a benchmark to measure these models in the past, and with the Graph MinCut problem. In both problems, our proposed approach improves performance and generalizes better compared to the standard planning budget.

## 1 Introduction

In the past few years, Deep Neural Networks (DNNs) produced substantial advances in a myriad of fields, from image classification and segmentation, through audio analysis and speech recognition, to generation of art and text with human-like accuracy. Despite these advances, a significant problem still remains: generalization to unseen inputs. Since a DNN is trained by solving an optimization task over a training set, the network often achieves lower performance on inputs that are out of distribution to it's training set. This can be attributed to sparse input distributions, outliers, edge cases and more. In order to improve generalization, DNNs are trained on ever increasing numbers of training samples. For example in NLP, dataset sizes can reach billions and trillions of tokens Kudugunta et al. (2023); Wang et al. (2023).

A potential solution to the issue of generalization lies in algorithms, which usually are able to solve a problem for all cases. Instead of learning to approximate a function, we learn to produce an algorithm; a series of steps, changing internal state, memory, or external interfaces, that eventually reach the desired outcome. We can hope that if a good algorithm is found, it will generalize to all cases by definition. This idea is often called Algorithmic Reasoning, and we use this name in this paper.

There are multiple examples of algorithmic reasoning, which can be implemented in an explicit or an implicit manner. In the explicit approach, the model's task is to output a description of the algorithm it has learned. Examples include AlphaTensor Fawzi et al. (2022), in which the model learns to find general matrix multiplication algorithms for various matrix sizes; code generation models such as Li et al. (2022), and Large Language Model (LLM) that are able to generate a piece of code solving a task described in free text Shinn et al. (2023).

In the implicit approach, the processor learns to output actions that work for a problem instance. For example in graph shortest path, the input is a specific graph $G$ and a query pair of source node and target node $(s, t)$. The correct output is the edges comprising the shortest path between $s$ and $t$. The processor's internal representation mimics a computer model, and its actions can be interpreted as instructions for processing memory or register values. However unlike the explicit approach, which outputs instructions that should work for all inputs, the implicit approach simply runs the learned algorithm on the given inputs. In order to run the algorithm, we must run the model. In this way, the model learns to perform the algorithm rather than describe it; the model's weights, internal

representation space, and architecture together comprise the learned algorithm. Examples include Zaremba & Sutskever (2016); Veličković et al. (2020); Kurach et al. (2016); Graves et al. (2014)

An important example of this approach is the Differentiable Neural Computer model Graves et al. (2016), which is the focus of this work. In brief, the DNC is a Recurrent Neural Network (RNN) based on a differentiable implementation of a Turing Machine, extending previous work Graves et al. (2014). Featuring an LSTM with a memory matrix, the DNC can model algorithms that interact with external memory, handling tasks like copying, sorting, and graph problems. A summary of the DNC architecture is provided in Appendix G.

In short, The DNC processes inputs by iteratively absorbing a sequence of vectors, storing them in memory, and executing memory operations for task-specific outputs. It has several addressing mechanisms which allow it to allocate new cells, update existing ones or lookup a specific cell by its content. Its operation spans three phases: input, planning, and answering. Initially, it receives input, then undergoes $p$ planning steps for processing—a number previously limited to zero or just 10 in more complex tasks—and finally produces the output in the answering phase.

We state and provide empirical evidence for two important claims:

1. Choosing an adaptive planning budget allows the model to learn a more general algorithm than it can with a constant budget, allowing it to generalize to input sizes much larger than seen during training.

2. Choosing an adaptive planning budget allows the model to learn better memory utilization, facilitating the use of larger memory for larger inputs.

Our findings show that a DNC trained with a constant $p$ faces limitations, most likely overfitting to heuristics and average case-attributes of the input. This issue persists unless it is trained with a substantially large $p$, which also has inherent limitations. In contrast, the adaptive model avoids these issues, demonstrating more robust performance without the constraints observed in the constant budget models. The paper is structured as follows: Section 2 overviews related work; Section 3 details our method and its complexity theory motivation; Section 4 presents experimental analysis supporting our first claim; Section 5 discusses larger constant budgets and evidence for our second claim; and Section 6 concludes the paper.

## 2 RELATED WORK

**Memory Augmented Neural Networks**  Memory-augmented neural networks (MANNs) are a class of neural network architectures that incorporate an external memory structure enabling it to store and access important information over long periods of time. The Differential Neural Computer (DNC) is one such network that has shown to be good at a variety of problems Graves et al. (2016); Rae et al. (2016). Since the DNC's introduction, many researchers have tried to improve this design. Franke et al. (2018) improved it specifically for question answering, while others have suggested changes to improve its overall performance. Csordás & Schmidhuber (2022) pointed out some issues with the DNC design and propose fixes. Yadav & Pasupa (2021) suggested separating the memory into key-value pairs. In another work Ofner & Kern (2021a), the authors tried to encourage loop structures in the learned algorithm by constraining the state-space of the controlling. There's also evidence that making the network more sparse can help with generalization and efficiency on bigger tasks Rae et al. (2016). Others propose new computational architectures such as the Neural Harvard Machine Tanneberg et al. (2020). None of these works specifically target the impact of the constant planning phase on the performance of DNC.

**Adaptive Computation Time**  Adaptive computation time is an important aspect of solving algorithmic tasks, as more complex instances naturally require more time to solve. Adaptive Computation Time (Graves (2017)) are RNNs that incorporates a neural unit to allow the model to dynamically change the number of computational steps. Bolukbasi et al. (2017) present Adaptive Early Exit Networks which allow the model to exit prematurely without going through the whole structure of layers. In the context of memory augmented neural networks, similar ideas have been proposed. Shen et al. (2017) intruduces an iterative reasoning mechanism, enabling the model to refine its predictions. Banino et al. (2020) utilizes the distance between attention weights attending the memory

as a measure of how many more memory accesses the model needs. They do so by incorporating an additional unit that outputs a halting probability, which is trained using reinforcement learning. These works, though very relevant to the claims in our paper, don't prove that adaptive computation times are a requirement. In our paper we directly deal with the large impact the duration of computation has on the model's performance. Allowing the model to choose its own computation time fits well with our claims in ths paper, though we show that even a naively chosen planning budget already improves the model's performance substantially, without the need to alter the training procedure or add new neural modules to the model.

## 3   ADAPTIVE PLANNING BUDGET

### 3.1   MOTIVATION

The DNC's flexibility to simulate an implicit algorithm, motivates using it to solve a wide variety of problems. However, by using a constant planning budget we limit the problems the model can solve to a specific complexity class. For example, a DNC with constant planning budget can model on-line algorithms with a constant latency. Clearly, this class of problems is quite restrictive. Since the adaptivity to input size is only present during the input phase, the model isn't free to do with it as it pleases, and has to either use it to save the raw input to memory, or use it to precompute some compression of the input, as in an on-line algorithm. For some problems that would require seeing the input as a whole the algorithm can effectively only dedicate a constant amount of time to actually solving the problem, which is very problematic when handling input sizes larger than that constant.

On the contrary, if this adaptive behavior is allowed during the planning phase, the DNC is freed from these limitations. Consequently, the number of algorithms which can't be expressed as on-line algorithms up to a constant latency is much larger than those that can. This other class is the one we're interested in, and includes e.g. Graph Shortest Path.

### 3.2   ADAPTIVE PLANNING BUDGET IMPROVES GENERALIZATION

**Adaptive Planning Budget**   Given an input $x$ with description length $|x| = n$, we set the planning budget to be a function of the input size, $p(n) \to \mathbb{N}$, which we call the *adaptive planning budget*. The DNC's runtime is therefore adaptively determined from the input's size. We now claim:

**Claim 1** *By using an adaptive planning budget during training, the model is able to learn a more general algorithm than it can with a constant planning budget, allowing it to generalize to larger inputs than seen during training.*

To validate our claim, we evaluate the model on inputs larger than those seen during training. If the model was able to learn a good representation of the input, and an abstract algorithm to process this representation, it should be able to perform well on larger inputs. Conversely, a model trained with the standard $p(n) = 10$ budget, as used in prior work, might resort to heuristics to solve the task on the training data within the confined planning time, leading in overfit and worse generalization.

We test this on two graph problems: Graph Shortest Path and Graph MinCut. See Appendix A for a full description of the problems. We train two models for each problem, with $p(n) = 10$ (baseline constant) and $p(n) = n$ (linear) planning budgets, four models in total. All models are given the same memory size of 200 cells, and are trained on the same data distribution of graphs with $|E| \leq 75$. To train the models, we use a similar curriculum learning approach to the one described in Graves et al. (2016). Finally, we test each model on graphs with different number of edges $|E|$, and estimate the models' accuracy. We use graphs with $|E| \in [5, 250]$ to test the models' ability to generalize to inputs much larger than seen during training. Results are in Figure 1.

As can be seen, the models with adaptive planning budgets achieve better generalization than the model with the standard constant budget, supporting our claim. In Section 5.2, we delve deeper, demonstrating that the training resources for these models are comparable. We also show that trying to replicate the linear model's performance with a larger constant budget results in costlier training and less efficient memory use. We note that the improvement is much more pronounced for the Shortest Path problem than the MinCut. We believe this is because a linear budget is not enough for the MinCut problem, who's best known algorithm has time complexity

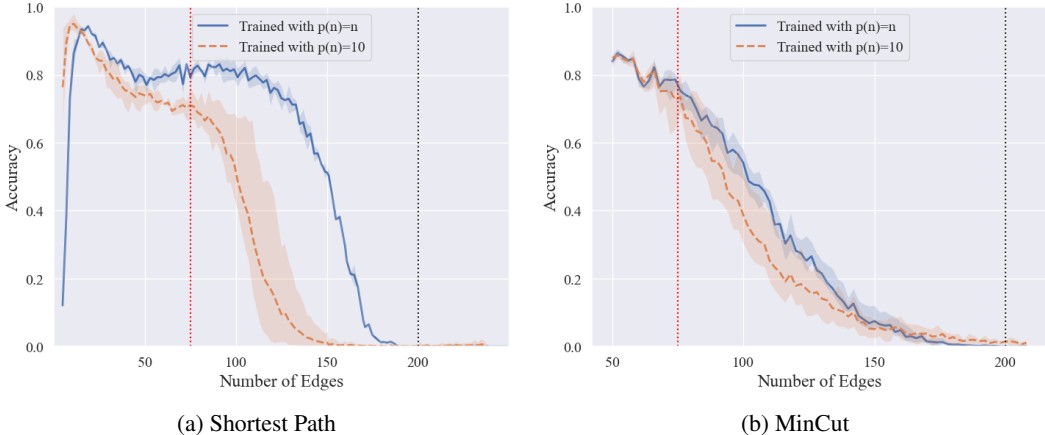

Figure 1: **Generalization to input sizes not seen during training** Performance of DNC models trained with a constant $p(n) = 10$ and a linear $p(n) = n$ planning budgets, on (a) Graph Shortest Path and (b) Minimum Cut problems. The accuracy is measured over graphs with different numbers of edges. Graphs seen during training have at most 75 edges for both problems, marked in **red**. The memory size is 200 cells, marked in **black**. using an adaptive budget achieves better generalization than the model with standard $p = 10$ budget, though the effect is more pronounced in the shortest path problem.

$\mathcal{O}\left(|V| \cdot |E| + |V|^2 \log(|V|)\right)$. Training with such a large computation budget was too prohibitive in terms of training time. Further discussion on the MinCut problem is presented in Appendix E. Full experimental setup can be found in Appendix A, and the intermediate results for each curriculum step are described in Appendix B.

### 3.3 ADAPTIVE PLANNING BUDGET IMPROVES MEMORY EFFICIENCY

As we seek to measure the DNC's generalization to larger inputs, we are immediately met with the finite size of its external memory. It is clear that just saving the input to memory is required for even simple tasks, such as Graph Shortest Path. This means we cannot run the DNC on inputs larger than its external memory. A proposed approach in Graves et al. (2016) is to use a larger external memory during test time. However subsequent studies indicated a decline in performance using this method. For example, Ofner & Kern (2021b) illustrated a decay in performance with an extended memory, and suggests that the controller learned a solution that is dependent on the memory size.

We propose a different explanation. Since the model was only allowed a small constant planning budget during training, it was forced to find an average-case algorithm that can be efficiently run, and whose internal representation of the data can be efficiently written to and read out of memory. This means that it attempted to minimize the number of memory operations required, encouraging the model to be less proficient at performing these operations. When we then try to extend the memory during inference, it simply fails to use it properly. Thus we can state the following claim, which we test in Section 5:

**Claim 2** *A DNC model trained with an adaptive planning budget learns to use its memory addresses in a more general way than it can with a constant planning budget. This enables it to (1) maintain memory retention over extended periods, (2) effectively manage a larger memory during inference, and (3) utilize memory more efficiently, scaling its memory usage in line with input size.*

We provide empirical evidence for the first aspect of the claim in Section 4.2, for the second in Section 5.1 and for the third in Section5.2.

## 4 EXPERIMENTS: IMPROVED GENERALIZATION

### 4.1 INFERENCE-TIME ADAPTIVE PLANNING BUDGET

During training, the model learns an implicit algorithm whose time complexity is unknown and may differ from the planning budget used in its training. Even if this learned algorithm truly generalizes, the planning budget we use in inference might simply be too short for the learned algorithm. Instead, by granting the algorithm a larger planning budget during inference, a general algorithm could achieve better generalization, even if it was found when training with a much smaller planning budget. This of course goes against our claim. To verify if this is the case, we compare the performance of a DNC model which is trained with a constant number of planning steps, and the same model when given a linear planning budget during inference.

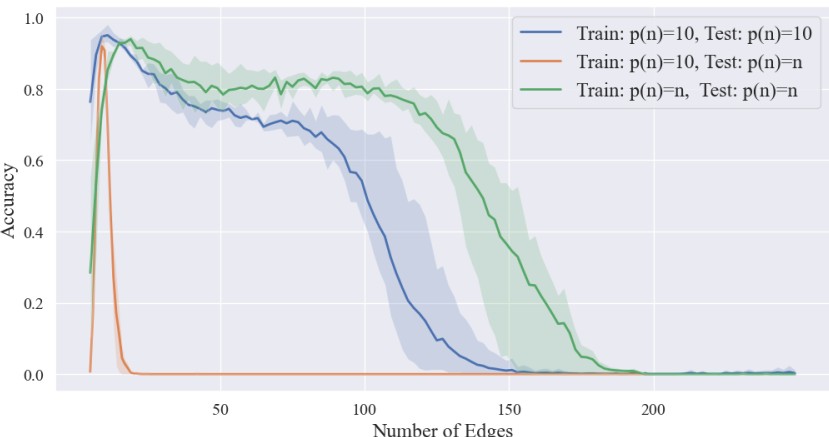

Figure 2: **DNC doesn't learn a generic algorithm when trained with** $p(n) = 10$ The generalization of a DNC model trained with a $p(n) = 10$ budget, evaluated with constant $p = 10$ and a linear $p(n) = n$. It is evident that giving the model a linear planning budget during inference deteriorates its performance severely, indicating that choosing the right planning budget during training is critical. We also show the generalization performance of a DNC with linear budget during training.

As we see in Figure 2, the model's performance is hugely impacted by the number of planning steps. Even a minor change to the planning budget impacts the model's performance significantly, and it doesn't generalize as well as the model trained with linear planning budget. This strengthens our claim that adaptive planning budgets are important during training. However, it is still possible that the learned algorithm could benefit from a different number of planning steps than the one used during training. We made a choice of $p(n) = n$ during our evaluation, which introduces a bias in our results. Instead, we could measure the model's required planning budget.

### 4.2 EMPIRICALLY DETERMINED PLANNING BUDGET

Instead of choosing our planning budget in advance, we can infer the optimal budget during inference by observing the model's performance. Let $A_n(p)$ be the model's accuracy on graphs with $|E| = n$ and a given number of planning steps $p$. For a specific value of $n$, we evaluate $A_n(p)$ for all $p \in [0, 300]$. Results for Shortest Path are shown in Figure 3 and for MinCut in Appendix E.

For the constant budget DNC, $A_n(p)$ is non-zero only near $p = 10$, indicating that it will not see any benefit from a different planning budget than the one used in training. In contrast, for the adaptive budget DNC, the function $A_n(p)$ shows a phase transition. With too few planning steps, the performance is low, but after some threshold value the performance jumps to a high level and remains there even if the number of planning steps is significantly increased. This phase transition value indicates a good choice for an empirically determined planning budget; using more planning steps is not very beneficial, and using less is detrimental. We mark this phase transition value as $p^\star(n)$, defined as the smallest $p$ for which accuracy exceeds 90% of its maximum value: $p^\star(n) = \arg\min \{p \mid A_n(p) > 0.9 \cdot \max\{A_n(p)\}\}$.

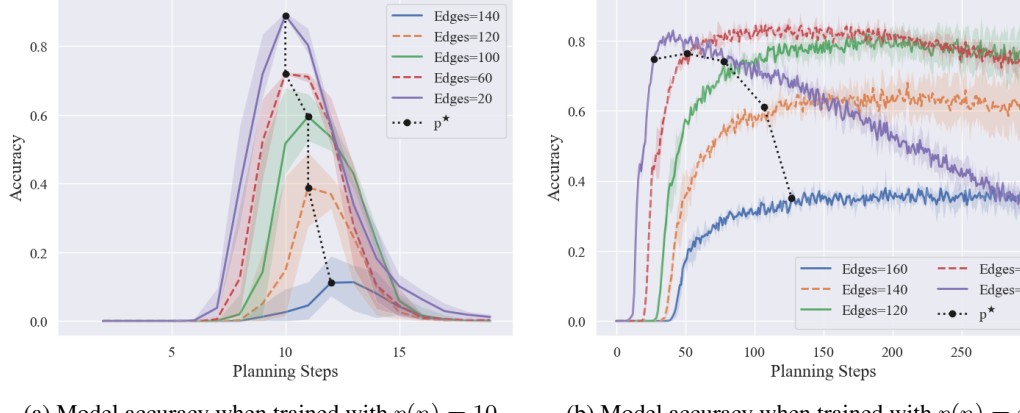

(a) Model accuracy when trained with $p(n) = 10$      (b) Model accuracy when trained with $p(n) = n$

Figure 3: $A_n(p)$ **for different input sizes** $n$**, Shortest Path** - Each colored line represents model accuracy over graphs of a chosen size, as a function of number of planning steps. Black dots denote the empirically determined planning budget $p^\star(n)$. The model trained with constant budget model only works when given $p = 10 \pm 5$, whereas the model trained with linear budget maintains stable accuracy across various $p$ values. This indicates the models learned truly different algorithms.

The phase transition in $A_n(p)$ can be understood as follows: For $p \leq p^\star(n)$, the model's accuracy could be improved if given more planning steps, suggesting that for some inputs the learned algorithm's runtime exceeds the planning budget. For all $p \geq p^\star(n)$, accuracy plateaus, indicating that the algorithm typically concludes before reaching $p$ planning steps, with the model then holding a 'finished' steady-state until the answering phase. This steady-state behavior suggests a truly generalized algorithm, contrasting with the less stable performance of the $p(n) = 10$ model. The adaptive model's ability to sustain its memory during this phase corroborates the second part of Claim 2, demonstrating its proficiency in preserving answers in memory without needing updates.

In Figure 4, we observe that for both problems, $p^\star(n) \approx p(n)$. This suggests that the models developed algorithms with runtimes matching their allotted training times, and that this runtime generalizes very well into graphs much larger than those seen during training. In the contrary, more planning steps don't enhance the constant-budget model's performance. Essentially, our results indicate the need to maintain the same planning budget during inference for optimal performance of the trained models. This strongly affirms Claim 1, that a constant budget DNC model struggles to learn general algorithms for complex tasks. Conversely, by simply allowing the DNC an adaptive planning budget such as $p(n) = n$, the DNC can learn a more "general" algorithm. Interestingly, while analyzing Figure 3b, we observed some inconsistencies in the linear budget models. Despite outperforming the $p = 10$ models, they exhibited less stable accuracy across different $p$ values. For further details ans figures see Appendix C.

## 5    Experiment: Improved Memory Utilization

### 5.1    Generalization to Larger Memory during Inference

As described in 3.3, the DNC's generalization is limited by the size of its external memory. Using a larger memory during inference was shown to perform well for simple tasks in Graves et al. (2016), but later work indicates this degrades performance for more complex tasks (Ofner & Kern (2021b)). We propose a modification of the memory extension, involving sharpening of the read weights for the content-based addressing mechanism. The content-based read weights are calculated in three steps. First, the controller produces a key vector $\boldsymbol{k} \in \mathbb{R}^C$. Then a similarity score is computed for each memory cell $\hat{\boldsymbol{c}} = \boldsymbol{M}^T \boldsymbol{k} \in \mathbb{R}^N$. Finally these scores are normalized using the softmax function $\boldsymbol{c} = \text{softmax}(\hat{\boldsymbol{c}})$. As a result, increasing the size of the external memory produces a smoother distribution, more spread out over the larger address space. We believe this leads to the observed degradation in performance.

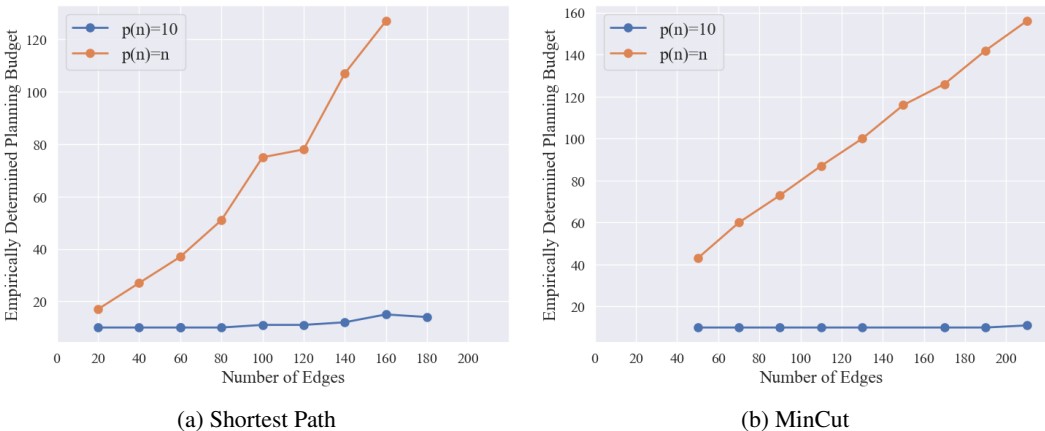

(a) Shortest Path                    (b) MinCut

Figure 4: **Empirically Determined Planning Budget** - $p^\star(n)$ measured empirically for different values of $n$ for models trained with $p = 10$ and $p(n) = n$.

To eliminate this problem we propose reweighting the content-based read weights by using a temperature parameter: $c = \mathrm{softmax}(\frac{1}{t}\hat{c})$. The optimal $t$ can be found through hyper parameter search. We found that a value of $t = 0.85$ works well for extending the memory to double its original size, and $t = 0.65$ allowed us to extend the memory five times, from 200 to 1000. We show results for the Shortest Path problem in Figure 5.

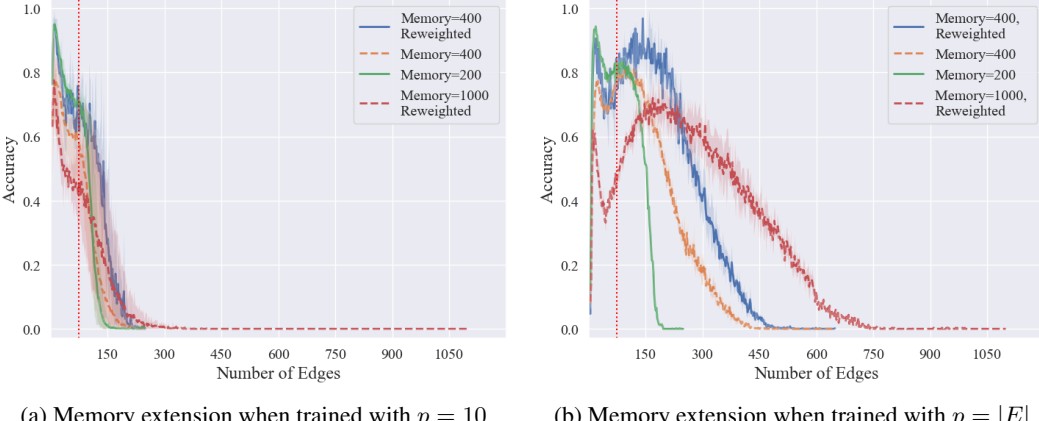

(a) Memory extension when trained with $p = 10$      (b) Memory extension when trained with $p = |E|$

Figure 5: **Reweighted Memory Extension, Shortest Path** - The extended memory significantly enhances generalization for the model trained with $p(n) = n$, reaching non-trivial accuracy on graphs 8 times larger than seen during training, given we extend its memory enough. On the other hand, the model trained with constant planning budget doesn't see extra generalization at all, even after reweighting the read weights. A **red** vertical line indicates the largest graph sizes seen during training.

The model trained with the standard constant planning budget, as used in previous research, shows no improvement with increased memory size, aligning with findings in Ofner & Kern (2021b). In contrast, the linear budget-trained model exhibits enhanced generalization with memory expansion. Simply doubling the memory from 200 to 400 cells improves performance on larger graphs, albeit with lower accuracy. Reweighting the 400-cell memory further boosts accuracy significantly. For example, the model's accuracy on training data graphs with 75 edges was 81%; this accuracy is surpassed on graphs with up to 215 edges, nearly 2.8 times larger, using reweighted 400-cell memory. With a 1000-cell memory, the model shows notable performance (over 15%) on graphs with 600 edges, 8 times larger than its training data.

This supports the second aspect of Claim 2, demonstrating that models with an adaptive planning budget can efficiently use a larger memory, even without a reweighting mechanism. The more abstract the algorithm learned by the model, the more effectively it can leverage increased memory, as this expansion alleviates memory size constraints. Additionally, while reweighting benefits both constant and adaptive models, the adaptive model shows superior performance gain when allowed to exploit the enlarged memory capacity, highlighting its superior memory utilization capabilities.

## 5.2 Adaptive Budget vs Larger Constant Budget

In this paper, we assess our adaptive planning budget against the established DNC baseline of $p(n) = 10$ from Graves et al. (2016), a standard in DNC research. While it might be suggested that simply increasing the number of constant steps could suffice, we argue that any constant budget inherently limits DNC. With a smaller constant budget, models may resort to heuristic solutions that work for training data but fail to generalize to longer inputs, a behavior we believe is observed in the standard $p(n) = 10$ model.

For a larger budget, two scenarios emerge. Firstly, the model could learn to use a constant empirically determined planning budget, likely adopting an online algorithm with fixed latency unsuitable for more complex challenges. On the other hand, it might successfully develop an adaptive algorithm but is restricted by the constant budget, inevitably facing input sizes beyond its capacity. However, this kind of adaptive algorithm that uses a large constant budget is much costlier, requiring more steps for smaller instances for training. Notably, our experiments have not demonstrated models trained with constant budget with an empirically determined budget that adapted to large inputs. More detailed discussion in Appendix F.

To understand DNC's behavior with larger constant budgets, we trained models with $p = 75$ and $p = 200$, assessing their generalization (Figure 6a) and training efficiency. To approximate the FLOPs used for training, we monitor the frequency of DNC model usage, termed as a sequence step. Each sequence step correlates with a certain constant number of FLOPs, allowing us to estimate the total computational cost. We then compare the number of sequence steps required for training to the model's generalization ability on graphs with 150 edges ($|E| = 150$), twice the size of the largest training sequence. The findings of this comparison are illustrated in Figure 6b. Figure 6 indicates that a $p(n) = 200$ budget is necessary to match the linear model's generalization, but this requires quadruple the computational resources. Intriguingly, the linear model's computational needs for training are similar to the $p(n) = 10$ model.

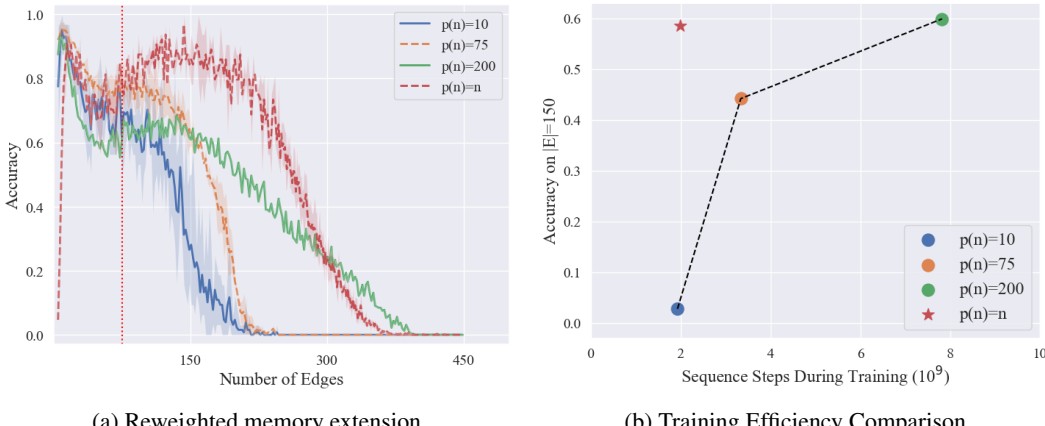

(a) Reweighted memory extension

(b) Training Efficiency Comparison

Figure 6: **Generalization And Training Efficiency of Larger Constant Budgets** - (a) Extending the memory to 400 significantly improves the generalization of the models with $p(n) = n$ and $p(n) = 200$. Models with smaller constant budgets $p(n) = 10, 75$ do not experience this improvement. (b) The adaptive model's training compute is similar to the baseline $p(n) = 10$, whereas $p(n) = 200$ demands four times the computational resources.

We delve deeper by comparing the memory utilization of the models that effectively generalize - $p(n) = n$ and $p(n) = 200$ - during their planning phases, aiming to determine if they've learned

similar algorithms. This also tests Claim 2, which posits that an adaptive planning budget in training enhances memory efficiency and scales its usage according to input size.Figure 7 demonstrates that the models learned different algorithms. The linear model's memory usage scales with the input size, whereas the constant step model overutilizes memory even for small inputs, supporting our claim. For additional information, figures, and MinCut results, refer to Appendix F.

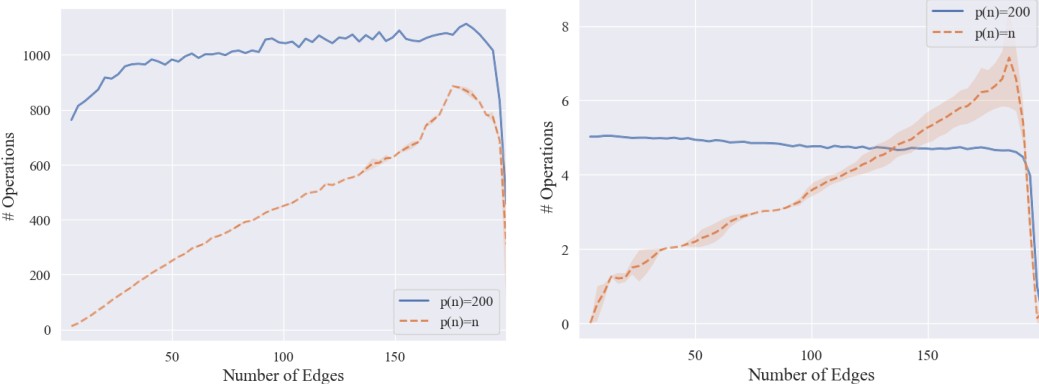

(a) Number of read cells during the planning phase  (b) Number of written cells during the planning phase

Figure 7: **Memory Efficiency** - A comparison of memory use during the planning phase reveals distinct algorithms behind the effective generalization of $p(n) = n$ and $p(n) = 200$ models. The adaptive model's memory utilization scales with input size, in contrast to the constant model, which consistently uses a higher but fixed number of cells, suggesting it may have learned an online algorithm with constant latency. The observed sudden drop at the end results from handling inputs exceeding the memory capacity of 200.

## 6   CONCLUSION AND FUTURE WORK

In this work we stated two important claims: first, a DNC model cannot be used to learn a general algorithm for complex problems if its planning budget is the standard constant budget; second, if we use an adaptive planning budget, the DNC is also able to learn to utilize its memory in a more robust and general way, allowing it to use a larger address space than seen during training. Both claims relate directly to algorithmic complexity, in terms of time complexity and memory complexity, and both claims help us find better DNC models that learn more general algorithms. In this paper we were able to show strong experimental evidence that simply choosing an adaptive planning budget greatly improves the performance of a DNC, and demonstrated it on two graph problems: Shortest Path and Minimum Cut.

The implications of our paper are probably not limited to the realm of DNCs, however. We can hypothesize that the runtime constraint we described, which can be lifted by using adaptive planning budgets, also applies to other models for algorithmic reasoning. These other methods must also find ways to enable sufficient time and memory complexity for their models, otherwise the resulting algorithms will be suboptimal.

In future work, we hope to explore usage of even larger planning budgets, which unfortunately is very prohibitive to do with the DNC model. We give preliminary results with a quadratic planning budget in Appendix D, which look very promising already. We also hope to take this approach to other models of algorithmic reasoning, such as code generation models etc.

**Reproducibility Statement**   This work follows the training procedures described in the original DNC paper, Graves et al. (2016), with the only modification of the number of planning steps used during training. For a thorough description of the training procedure, we direct the reader to Appendix A. For reproduction of the memory extension experiments, all details required are presented in Section 5.

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

# A   APPENDIX A - TRAINING SETUP

In this work, we use two tasks to demonstrate our claims:

1. **Graph shortest Path Task** This task has been used previously to evaluate DNC. In this task, the model sequentially receives a description of the graph $G(V, E)$ as its set of edges over $|E|$ steps, a query $(s, t)$ where $s$ is the source node and $t$ is the target node, and its output is a set of ordered edges describing the shortest path from $s$ to $t$.

2. **MinCut Task** This task has not been used before as a an algorithmic task to the best of our knowledge, the model also receives a description of a connected graph $G(V, E)$ as its set of edges over $|E|$ steps, but no query. The output is a set of edges describing the global minimum cut of the graph. As a reminder, a cut is a set of edges that once removed from the graph it will no longer be connected. A minimum cut is a cut with the minimal number of edges.

## A.1   DATA GENERATION

For the training process, we adopted a curriculum-based approach, training the model on increasingly larger graphs and more complex queries. This meant parameterizing the dataset by graph size. Every 1000 training steps, the model is evaluated on the current lesson. If its accuracy surpasses 80%, we move training to the next lesson. The models trained on the final lesson for $100K$ steps overall, with the exception of the model with $p = |E|^2$ which trained for $3K$ steps due to its high training time.

### A.1.1   SHORTEST PATH TASK GRAPH GENERATION

Each curriculum lesson is parameterized by the number of nodes $[n_1, n_2]$, average degree $[d_1, d_2]$, and path length $[p_1, p_2]$. The training graph are sampled uniformly from the set of all graphs with $n$ nodes and $m$ edges where $n$ is uniformly sampled from $[n_1, n_2]$, and the number of edges $m$ is uniformly sampled from $\left[ \lfloor \frac{N \cdot d_1}{2} \rfloor, \lfloor \frac{N \cdot d_2}{2} \rfloor \right]$.

### A.1.2   MINCUT TASK GRAPH GENERATION

Each curriculum lesson in this task is parameterized by the number of nodes $[n_1, n_2]$, clusters $[C_1, C_2]$, and cut size $[c_1, c_2]$. The generator samples graphs as follows:

1. Split the $n$ nodes into $k$ disjoint groups, denoted as $C_1, ..., C_k$. Each group contains at least $c + 2$ nodes to ensure a non-trivial minimum cut.

2. For each group $C_i$ (where $1 \leq i \leq k$), randomly sample a graph from the space of all graphs with $|C_i|$ nodes and a minimum degree of at least $c + 1$.

3. Randomly add $c$ edges to connect $C_1$ to the rest of the nodes.

4. If there are more than two groups ($k > 2$), add edges between different clusters to ensure that each cluster is connected to at least $c$ edges, or $c + 1$ if a unique minimum cut is required.

For the purpose of this work, we used a constant $[C_1, C_2] = [2, 3]$, and constrained the maximum number of edges in the graph overall by adding an additional parameret max degree $[d_1, d_2]$

### A.1.3   TARGET CONSISTENCY

For the Shortest Path Task, given a single graph and a query, multiple valid shortest paths could exist. This creates a possibility of the sample input sample having different targets throughout the training, creating ambiguity within the training samples. To overcome this problem, we used graphs with a unique shortest path for training purposes. Each sampled graph was modified to ensure a unique solution. This was done by computing all possible shortest paths using breadth-first search and removing edges that disconnect all but one of the shortest paths. The modified graphs were used for training purposes only. For model evaluation, the generated graphs were used as is, and the model was correct on a query if its prediction matched any of the valid shortest paths. Similarly,

for the MinCut Task, we face a similar problem, and trained similarly by generating graphs with a unique minimum cut for training, and use general graphs for evaluation purposes.

### A.1.4 GRAPH REPRESENTATION

For a sampled graph $G(V, E)$, each node in the graph is assigned a unique label sampled from $[1, N_{max}]$, where $N_{max}$ is the maximum number of nodes that the model supports. Each node label was encoded as a one-hot vector of size $N_{max}$, making the size of each input vector in the sequence $2 \cdot N_{max} + 2$, including the edge as well as the $< eoi >, < ans >$ tokens. The final graph description consists of the set of edges $x \in \mathbb{R}^{|E| \times 2 \cdot (N_{max}+2)}$.

### A.1.5 TRAINING

As we explained in section G, the input sequence is divided into several distinct phases: graph description, query, planning and answer phases. For a sampled graph $G(V, E)$ with $|V| = n$ nodes and $|E| = m$ edges, a sampled query $q$, and a set of a predefined number of planning steps $p$, the answer phase starts at $t_a = (m + 1 + p)$. To train the model, the model's output is considered in the cost function solely during the answer phase. During this phase, the model initially receives an answer cue indicating the start of this phase, and its output is utilized as feedback for the next step until the termination token is received.

The model has $2 \cdot N_{max}$ output nodes, corresponding to 2 softmax distributions over the two labels describing a single edge. Consequently, the log probability of correctly predicting the edge tuple is the sum of the log probabilities of correctly classifying each of the nodes.

For clarity, in the next section we denote the policy that the model learns over the actions $a \in \mathcal{A}$ by $\pi(a|s)$, where $s \in \mathcal{S}$ is the current state of the model. Additionally, we will refer to the correct answer sequence by $y = [y_1; ...; y_T]$, where $T - 1$ is the length of the shortest path. Lastly, the output of the model at each time step is denoted by $o_t = [o_t^1; o_t^2]$.

The cross-entropy loss corresponding to a single time step in the answer phase is:

$$\ell(o_t, y_t) = - \sum_{i=1}^{2} \log \left[ \Pr(y_t | o_t^i) \right]$$

And the overall loss over the whole input sequence:

$$\mathcal{L}(o, y) = \sum_{t=0}^{T} \ell(o_{t+t_a}, y_t)$$

In addition, we used teacher forcing to demonstrate optimal behaviour. This is commonly used in training recurrent neural networks that use the output from previous time steps as input to the model. Since during the answer phase the model's prediction at time $t$ is a function of $o_{t-1}$, the teacher forcing provides the model with the correct answer instead of its own prediction allowing it to learn the next prediction based on the correct history. This helps in the early time steps when the model has not yet converged. Formally, the current state during the answer phase $t > t_a$ is a function of the output of the model in the previous step $s_t = f(o_{t-1})$, and the next output is calculated as $o_t = \pi(\cdot|s_t)$. When the model is trained using teacher forcing only, the current loss is calculated as a function of the correct prediction in the previous step. Overall the loss is calculated as follows:

$$\mathcal{L}(\hat{o}, y) = \sum_{t=0}^{T} \ell(\hat{o}_{t+t_a}, y_t)]$$

where:

$$\hat{o}_t = \begin{cases} o_t & \text{if } t \leq t_a \\ \pi(\cdot|f(y_{t-1})) & \text{otherwise} \end{cases}$$

In practice, we followed Graves et al. (2016), and used a mixed training policy to guide the answer phase, by sampling from the optimal policy with probability $\beta$ and from the network prediction with probability $1 - \beta$.

## B APPENDIX B - CURRICULUM & INTERMEDIATE RESULTS

| Lesson | Nodes | Average Degree | Path Length | $DNC_{10}$ | $DNC_{|E|}$ | $DNC_{|E|^2}$ |
|--------|---------|----------------|-------------|------------|-------------|----------------|
| 1 | (5,10) | (1,2) | 2 | 8.1 | 0.0 | 2.1 |
| 2 | (5,20) | (1,2) | 2 | 19.3 | 3.9 | 10.4 |
| 3 | (10,20) | (1,2) | 2 | 22.6 | 5.9 | 13.2 |
| 4 | (10,20) | (1,2) | (2,3) | 30.7 | 12.1 | 27.8 |
| 5 | (10,20) | (1,2) | (2,3) | 42.0 | 33.8 | 51.5 |
| 6 | (10,20) | (1,3) | (2,3) | 50.3 | 45.1 | 53.9 |
| 7 | (10,20) | (2,3) | (2,4) | 60.8 | 52.7 | 50.6 |
| 8 | (10,20) | (2,3) | (2,4) | 66.6 | 66.2 | 67.7 |
| 9 | (10,25) | (2,4) | (2,4) | 72.8 | 73.3 | 67.1 |
| 10 | (10,25) | (2,4) | (2,5) | 76.0 | 77.1 | 68.0 |
| 11 | (15,25) | (2,4) | (2,5) | 78.9 | 79.7 | 70.3 |
| 12 | (15,25) | (2,5) | (2,5) | 79.3 | 79.4 | 71.6 |
| 13 | (20,25) | (2,5) | (2,5) | 81.1 | 81.3 | 71.3 |
| 14 | (20,25) | (2,6) | (2,5) | 90.4 | 92.2 | 72.2 |

Table 1: **Shortest Path Task curriculum and intermediate results** - parenthesis represent ranges (minimum value, maximum value). Evaluation of the models with different planning budgets is performed at the end of every lesson, and tested on graphs sampled from the final lesson in the curriculum

| Lesson | Nodes | Cut Size | Max Degree Per Cluster | $DNC_{10}$ | $DNC_{|E|}$ |
|--------|---------|----------|------------------------|------------|-------------|
| 1 | (10,15) | (1,1) | 3 | 0.0 | 0.0 |
| 2 | (10,15) | (2,3) | 5 | 1.8 | 1.9 |
| 3 | (15,20) | (2,3) | 5 | 67.3 | 64.1 |
| 4 | (15,20) | (2,4) | 6 | 69.8 | 66.5 |
| 5 | (20,25) | (2,4) | 6 | 80.7 | 80.9 |

Table 2: **MinCut Task curriculum and intermediate results** - Evaluation of models trained with different planning budgets on the final lesson in the curriculum during the training process.

## C APPENDIX C - LINEAR BUDGET FAILURE CASES

Training DNC models with linear planning budget allows them to learn algorithms that successfully generalize to larger input sizes, and learned to utilize memory, as presented in Section 4.2. However, some models trained in the exact same manner turned out to be failure cases who behaved similarly to constant budget models. As can be seen in Figure 8a, these models are sensitive to $p$ changes, and do not maintain a steady state when tested on larger inputs. As expected, when these models are given a larger memory during inference they fail to utilize it, as can be seen in Figure 8c. However, unlike constant budget models, they did learn some linear time algorithm, as evident when considering their empirically determined planning budget, presented in Figure 8b. This implies that the models learned a linear time algorithm but did not learn to utilize memory, even when given the resources and time that would allow it.

As a result, we considered it beneficial to present these distinct failure cases separately, rather than average their results with the successfully generalizing models, which may be misleading. Notably, the only variable changed causing these two distinct groups is the choice of seed used to set the pseudorandom elements of the experiment. Consequently, we could attribute this instability of the training procedure to the lack of hyperparameter tuning in both model parameter choices, learning rate and more. It's possible that after tuning these hyperparameters the training process will stabalize, but we didn't verify this possibility.

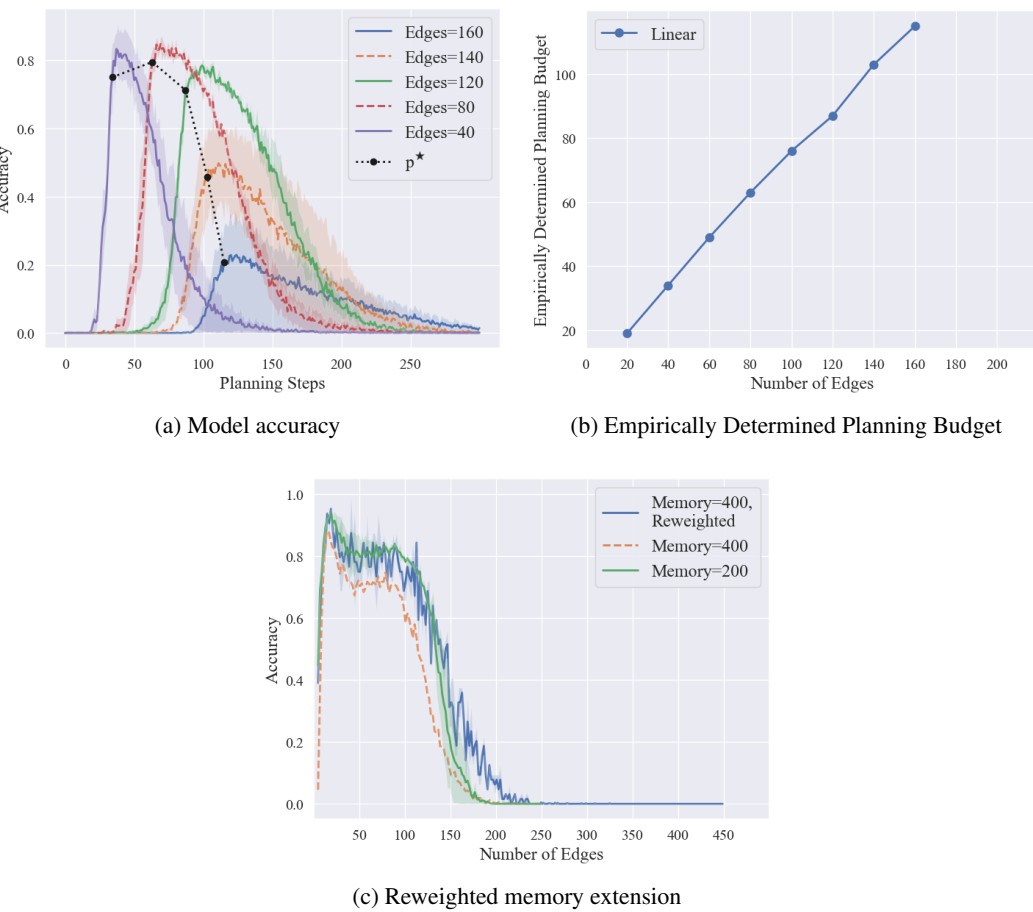

(a) Model accuracy

(b) Empirically Determined Planning Budget

(c) Reweighted memory extension

Figure 8: **Empirical Results for "failure cases" when training with linear planning budget, Shortest Path.** While these models learned a linear time algorithm, they did not learn to utilize memory, resulting in a generalization ability similar to a model constrained with a constant planning budget.

## D    APPENDIX D - QUADRATIC PLANNING BUDGET RESULTS

We also supply initial results with a quadratic planning budget $p = n^2$ trained on Graph Shortest Path. These models require a very long training time and as a result, we had to cut short the experiment after a mere 3% of the training process. We stopped the model after 3K steps in the last lesson, while for the other budgets we performed 100K.

Even with this extreme setback, a model that applies a quadratic planning budget reaches higher accuracy for large inputs, compared to models that use smaller budgets, as can be seen in Figure 9c. Furthermore, it appears that although the model uses a quadratic number of planning steps, it reaches the final steady state described in Section 4.2 very quickly. A nearly constant number of planning steps is needed, as can be seen in figure 9a,9b. As Shortest Path can be solved in linear time, we hypothesize this model was able to learn a very efficient representation space while still learning to use it's memory. This could be explained by the very large duration of the planning phase: for most input sizes during training, the model really doesn't need most of the planning phase it is give. This forces it to become very good at holding the steady state it reaches after finding the final answer, which perhaps allows it to then further optimize its representation. This in turn reduces its actual runtime considerably.

This result raises a question regarding constant planning budget - perhaps we simply didn't give a constant that is high enough to allow generalization to any size, although this is unlikely from a computational point of view. We trained a constant model with $p(n) = 25$ and observed the same behaviour as of $p(n) = 10$), so the answer is no. Another question is "what is the largest input this model can generalize to", which we show preliminary results in Figure 9d. This shows even better generalization performance than we saw with the linear budget model, which is very promising.

Unfortunately the very long runtimes become prohibitive to train such a model. This points at a limit of solving such problems using DNCs - complex problems will require a large amount of planning steps to train.

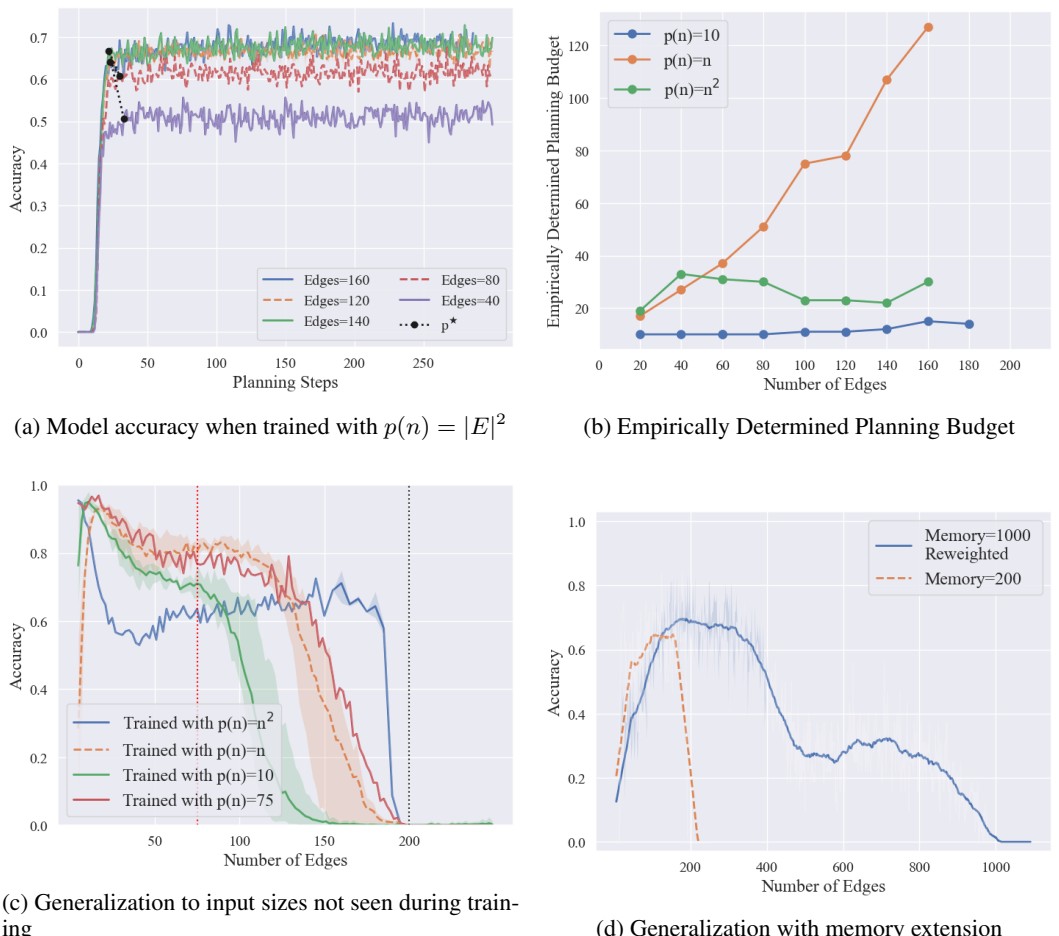

(a) Model accuracy when trained with $p(n) = |E|^2$

(b) Empirically Determined Planning Budget

(c) Generalization to input sizes not seen during training

(d) Generalization with memory extension

Figure 9: **Empirical Results for the partially trained model that trains with a quadratic planning budget.** It is evident that even with 3% of training time, the model surpasses smaller budgets when generalizing to larger input sizes.

# E   APPENDIX E - MINCUT SUPPLEMENTARY FIGURES

We supply additional results for solving the MinCut task with an adaptive planning budget. As can be seen in Figure 10, we again prove models trained with linear planning budget learn algorithms that can generalize to larger inputs, while constant planning budget do not. Unlike Shortest Path, when performing our test of memory extension the algorithm did not generalize well to larger inputs. The results we show do prove that the model learns a different algorithm, namely a linear time algorithm. It improves it's performance, but not as much as we saw in the shortest path. We believe this is because the MinCut requires $O(|V||E|)$ time to solve Stoer & Wagner (1997), while only linear planning budget was used. Unfortunately, like in Appendix D, training a model with a planning budget of size $|V||E|$ proved prohibitive and we had to stop it before it reached its final lesson in the curriculum, preventing us from presenting even preliminary results with this larger budget.

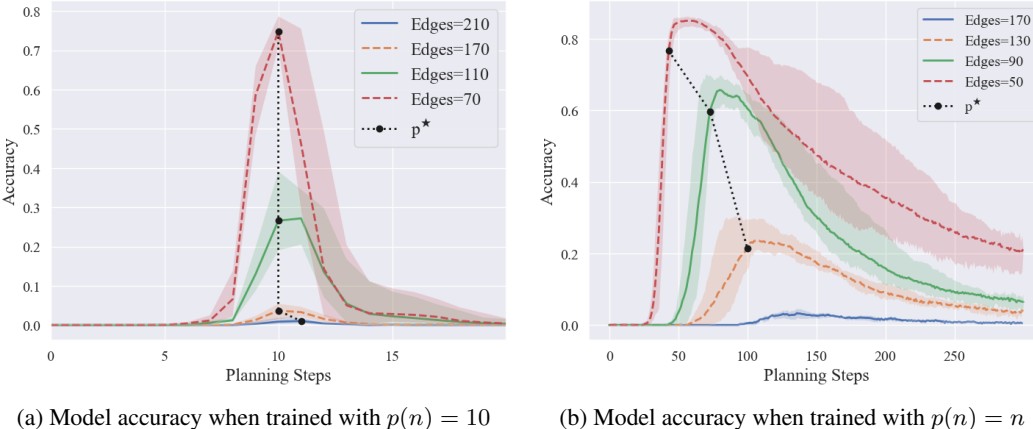

(a) Model accuracy when trained with $p(n) = 10$

(b) Model accuracy when trained with $p(n) = n$

Figure 10: $A_n(p)$ **for different input sizes** $n$, **MinCut** - The model trained with constant budget only works when given $p = 10 \pm 5$, whereas the model trained with linear budget maintains higher accuracy across various $p$ values, even for this harder problem.

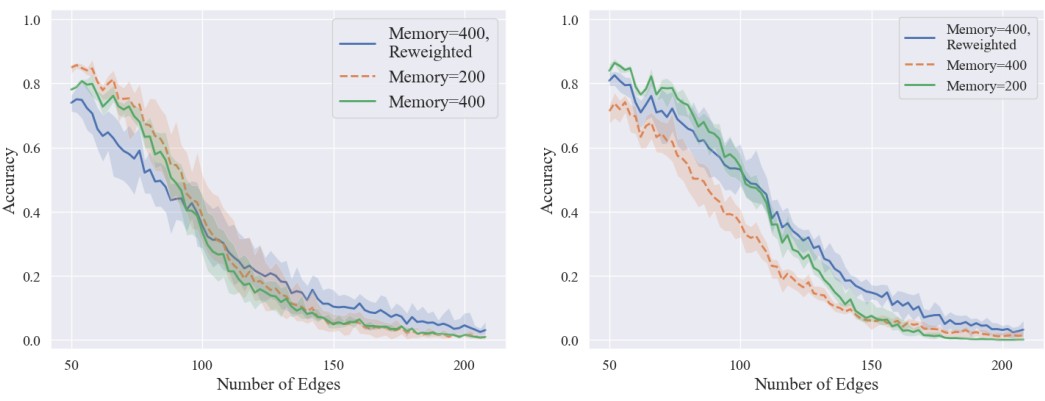

(a) Reweighted memory extension for model trained with $p(n) = 10$, on MinCut.

(b) Reweighted memory extension for model trained with a $p(n) = |E|$, on MinCut.

Figure 11: **Reweighted Memory Extension on MinCut** - On this harder problem it appears that the models did not learn to utilize memory, as linear time budget might not be sufficient.

# F  APPENDIX F - LARGER CONSTANT PLANNING BUDGETS

## F.1  GENERALIZATION TO LARGER INPUTS

Throughout the paper, the main constant baseline was $p(n) = 10$, as it was suggested in Graves et al. (2016) and adopted later on in following research. However, comparing our adaptive method with constant p=10 is problematic. A better comparison would be first between models that use a different constant. Then, comparing model with a large planning budget against a model with an adaptive budget would emphasize the strength of adaptivity.

So, in this appendix we provide additional empirical evidence for these comparisons, initially presented in Section 5.2. We compare our proposed adaptive budget to the constant one we train additional model with larger constant budgets: $p(n) = 75$, the largest graph size within training, and $p(n) = 200$. The generalization of these models can be seen in Figure 12.

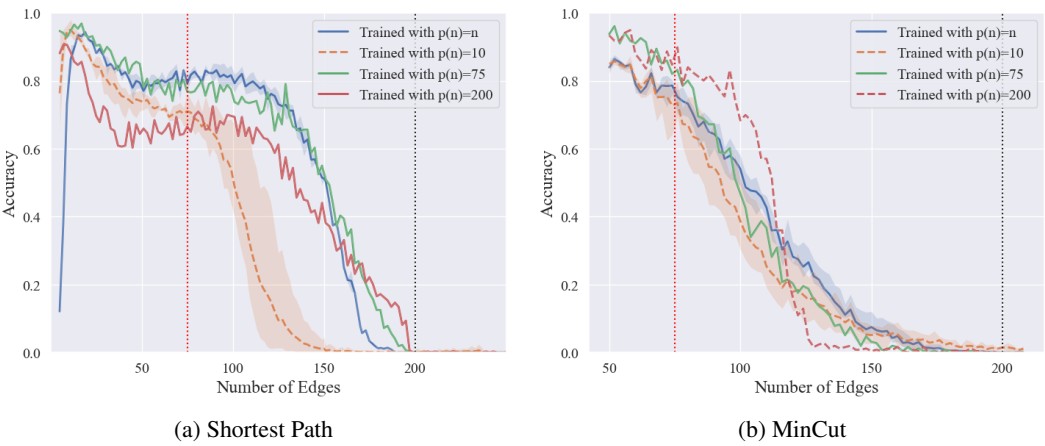

(a) Shortest Path  (b) MinCut

Figure 12: **Generalization to input sizes not seen during training** Performance of DNC models trained with two larger planning budgets, on (a) Graph Shortest Path and (b) Minimum Cut problems. The accuracy is measured over graphs with different numbers of edges. Graphs seen during training have at most 75 edges for both problems, marked in **red**. The memory size is 200 cells, marked in **black**. Increasing the constant budget achieves better generalization than the model with standard $p = 10$ budget, though the effect is more pronounced in the shortest path problem.

As we can see, a larger constant planning budget improves the generalization of the baseline with $p = 10$ model, bringing its generalization ability (with the original memory size of 200 cells) up to par with the adaptive linear model. However, this comes at the cost of requiring more training computation for training as shown in Figures 6b, 14a. This displays the ability of the adaptive model to learn a generalizing algorithm, while still matching the computation resources of the baseline $p = 10$ model.

## F.2  MEMORY UTILIZATION OF MINCUT

We supply the results of extending the memory for the Mincut problem in Figure 13. We note that all models don't perform well on the MinCut problem, unlike the shortest path, and we believe this is because the MinCut requires $O(|V||E|)$ time to solve Stoer & Wagner (1997), while only constant or linear planning budgets were used in our experiments.

## F.3  TRAINING EFFICIENCY OF MINCUT

We supply the results for the computational cost for MinCut in Figure 14.

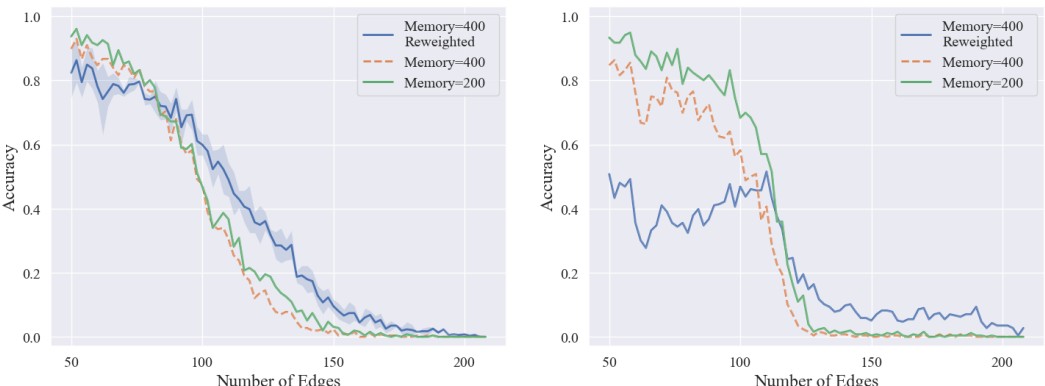

(a) Memory extension when trained with $p(n) = 75$    (b) Memory extension when trained with $p(n) = 200$

Figure 13: **Reweighted Memory Extension on MinCut** - On this harder problem it appears that the models did not learn to utilize memory, as linear time budget might not be sufficient.

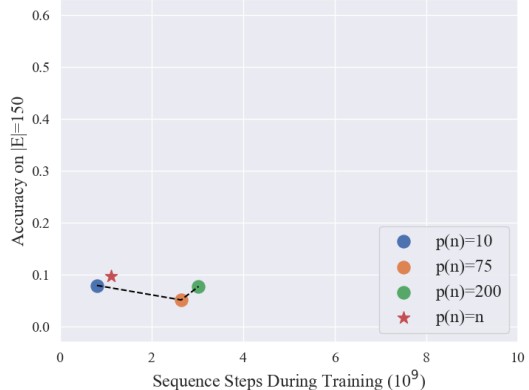

(a) MinCut - Training Efficiency Comparison

Figure 14: **Training Efficiency of Different Planning Budget** Number of sequence steps required to train the different models, on the MinCut problem. The accuracy is measured over graphs with 150 edges, twice the size of the largest training instance. The computations used to train the adaptive model match those of the constant model.

### F.4 EMPIRICALLY DETERMINED PLANNING BUDGET

Figures 15 and 16 summarize the results in line with Section 4.2, for the models trained with larger constant budgets $p = 75, 200$, for both Shortest Path and MinCut problems.

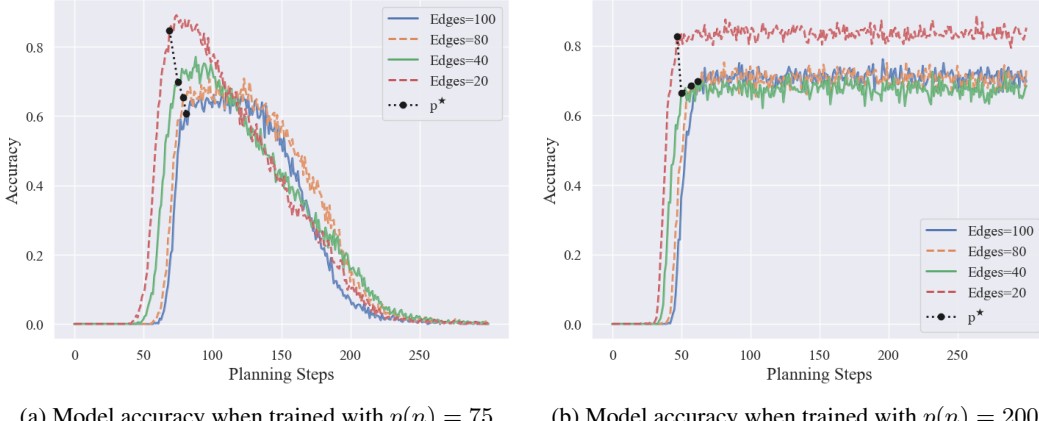

(a) Model accuracy when trained with $p(n) = 75$      (b) Model accuracy when trained with $p(n) = 200$

Figure 15: $A_n(p)$ **for different input sizes** $n$**, Shortest Path** - Each colored line represents model accuracy over graphs of a chosen size, as a function of number of planning steps. Black dots denote the empirically determined planning budget $p^\star(n)$.

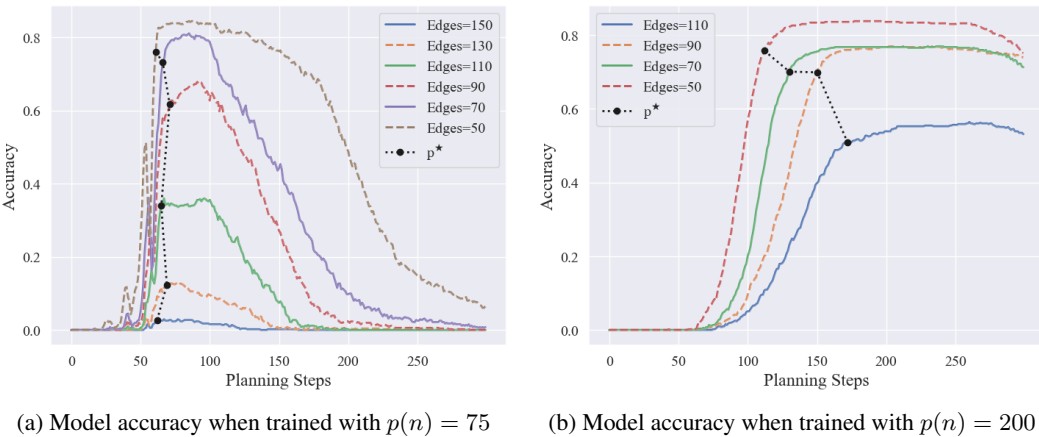

(a) Model accuracy when trained with $p(n) = 75$      (b) Model accuracy when trained with $p(n) = 200$

Figure 16: $A_n(p)$ **for different input sizes** $n$**, MinCut** - The larger the constant budget, the more flexible the model is with changing the number of planning steps provided.

Figure 17 shows the empirically determined budget for all models. In Figure 17a, We observe that all the constant models, learned a constant empirically determined budget. Notably, the model trained with $p(n) = 200$ has an empirically determined budget that is lower than the one trained with $p(n) = 75$. Additionally, Figure 6a indicates that the $p(n) = 200$ model is the only one to generalize comparably to the linear model, suggesting it may have learned an online algorithm with constant latency.

In Figure 17b, there's a striking contrast in the empirically determined budgets for $p = 75$ and $p = 200$. The $p = 75$ model maintains a steady empirically determined budget, similar to other constant models. In contrast, the $p = 200$ model displays adaptive behavior up to a certain input size. However, around $|E| = 130$, several notable observations are made:

- The empirically determined budget exceeds 200, the original training budget of this model.

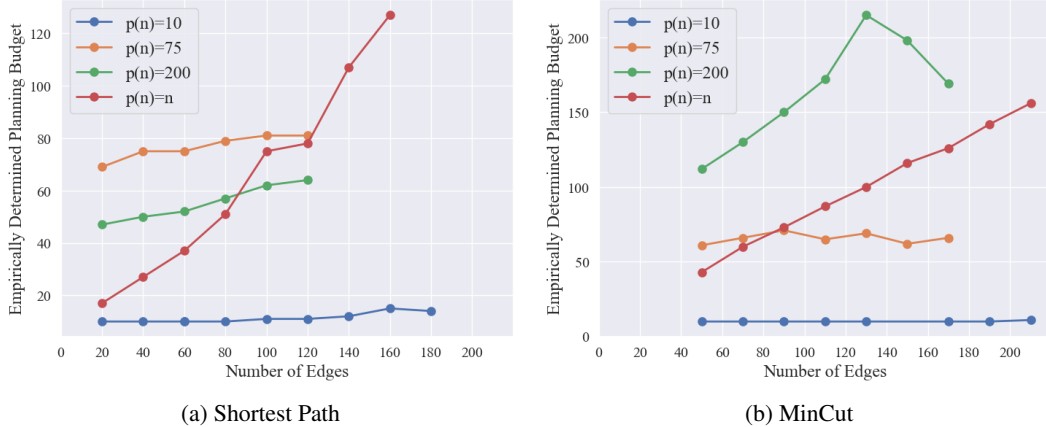

(a) Shortest Path             (b) MinCut

Figure 17: **Empirically Determined Planning Budget** - $p^\star(n)$ measured empirically for different values of $n$ for models trained with $p = 75$ and $p(n) = 200$.

- This budget, which had been increasing, begins to decrease, indicating that for inputs larger than $|E| = 130$, additional planning steps don't enhance performance.
- Concurrently, Figure 13b shows a significant performance drop around this input size, also revealing that increasing memory size doesn't substantially improve performance.

Overall, these findings imply that the model may have successfully learned an adaptive algorithm within its training constraints, but it hits a limit in handling larger inputs

## G APPENDIX G - DNC RECAP

The Differentiable Neural Computer (DNC) Graves et al. (2016) is a memory-augmented neural network, based on a recurrent neural representation of a Turing Machine. A DNC consists of a controller coupled with an external memory $M \in \mathbb{R}^{N \times C}$, where $N$ is the number of memory cells and $C$ is the size of a memory cell. The external memory can be accessed by the controller through different addressing mechanism, allowing the model to write to unused memory cells, update cells, and lookup specific cells based on their content. The external memory allows the DNC to be a general problem solver, and it was shown to work successfully on a wide range of tasks such as sorting, question answering and more.

The controller learns to interact with the memory using $m$ fully differentiable read heads and a single write head, allowing the model to learn to utilize $M$ through an end-to-end training process. Each read head $R^i$ accesses the memory at every timestep $t$ by generating weights over the address space $\boldsymbol{w}_t^{r,i} \in \mathbb{R}^N$, with the read value being $\mathrm{r}_t^i = \boldsymbol{M}_t^\top \boldsymbol{w}_t^{r,i}$. Similarly the write head $W$ updates the memory through a generated write distribution $\boldsymbol{w}_t^w \in \mathbb{R}^N$.

The controller of DNC acts as a state machine, managing memory access through generated signals. Its input is a concatenation of the input vector at time step $t$ denoted $x_t$, as well as the $m$ values read from the memory at the previous timestep, $\boldsymbol{v}_{t-1}^i$. The output of the controller is mapped into two vectors: a control vector $\xi_t$ used to control the memory operations, and an output vector $\nu_t$ used to generate the final output $o_t$.

The process of generating an answer sequence by DNC can be divided into multiple phases:

1. **Description Phase** A description of the problem instance is presented to the network sequentially. e.g. in Graph Shortest Path, the description of an input graph $G(V, E)$ can be the set of its edges.

2. **Query Phase** An end-of-input token denoted $\langle \text{eoi} \rangle$, followed by an optional query. In Graph Shortest Path, the query is the source and target nodes $(s, t)$.

3. **Planning Phase** For $p \geq 0$ time steps, a zero-vector input is provided, allowing the model time to access and update its state and external memory.

4. **Answer Phase** Initiated by an answer token denoted $\langle ans \rangle$, the model outputs the answer sequence $y$.

The $\langle eoi \rangle$, $\langle ans \rangle$ tokens are represented as binary channels, concatenated to the input vector. The $\langle ans \rangle$ channel remains active throughout the answer phase, while the $\langle eoi \rangle$ is presented to the model only once, at the onset of the query phase.

