# OpenReview forum: "DNCs require more planning steps"
_ICLR.cc/2024/Conference — Submitted to ICLR 2024_

### Official Review · Reviewer_Mmrw · 2023-10-23

**Soundness:** 1 poor
**Presentation:** 2 fair
**Contribution:** 2 fair
**Rating:** 5
**Confidence:** 4

**Summary:**

The authors investigate the impact of the planning budget on the performance of Differentiable Neural Computers (DNCs) in two algorithmic tasks: shortest path and mincut. They assert that an adaptive computation budget results in improved generalization for algorithmic tasks. Additionally, they explore the effects of increasing both the budget and the DNC memory during inference.

**Strengths:**

*  The topic is relevant and the paper raises interesting questions.
*  Adaptive computation budget is practically useful as well as interesting from the ML perspective (out of distribution generalisation, using the model in different regime compared to training).
* I like the idea of adjusting the softmax temperature when increasing the memory size.

**Weaknesses:**

* I find the story incoherent. Often this might be because the story is built around the computation budget, whereas the underlying story, in my opinion, is about out-of-distribution input size generalisation and using planning budget and memory increase for that. I think this leads to some unnecessary complications and breaks the flow of the argument, e.g. as with Section 5 resulting from 3.2.1. and having a two-page section in between.
* Some of the important topics are glanced over whereas more relevant parts could have been explored in more details. For the readers, unfamiliar with DNCs, this paper would be hard to grasp. I think, more time should be spent on Section 2, including an illustration would greatly help. This can be done by reducing Section 3.1. to a single paragraph describing the general inspiration.
* The paper is sometimes too loose when it comes to the results interpretation and some phrasings. I will put more details on this in the 'questions' subsection of the review, but here are some examples:
  - Section 3.2 is called that 'adaptive planning budget improves generalisation'. Figure 1 shows that training with budget = n is better than training with budget = 10. However, we don't know what will happen when we train with n=20 which is fixed, but more than the original n=10.
  - Section 3.2 proposes an explanation that is not scrutinised in the experimental section: "Since the model was only allowed a constant planning budget during training, it was forced to find an average-case algorithm that can be efficiently run, and whose internal representation of the data can be efficiently written to and read out of memory." To clarify things, I find this hypothesis extremely interesting and worth exploring. But it has to be tested in some way.
  - I find phrases like 'learning a more general algorithm' to be quite vague and misleading. Furthermore, in my opinion they make it much harder to read the paper as it would be if the reasoning was just about generalisation properties of the model. For example, instead of 'learning a more general algorithm' on can say 'generalise to a wider range of input sizes' or smth. I believe reasoning about the effect of a particular experiment from the perspective of out-of-distribution generalisation would be easier to interpret and reason about.

**Questions:**

I left some major suggestions in the 'weaknesses' section above. I will put concrete major questions below and will have a separate subsection for smaller comments/nits.

Major Qs:
* Could you provide supporting evidence that adaptive planning budget improves generalisation and it's not just about the max number of steps the model was allowed to use during training? What if we pick the constant number of steps that is the upper bound on the number you used for the adaptive computation curve on Figure 1?
* Could you explain the difference between your claim 1 and claim 2 stated in the intro? As far as I understand, claim 1 means "adaptive planning budget -> better generalisation", claim 2 means: "adaptive planning budget -> better use of memory -> better generalisation". Is claim 2 a more refined version of claim 1, or I'm missing something?
* Do you think it's possible that Figure 4 result is an artefact of giving the model n planning steps during training? It would be extremely curious to see a similar line for the model trained with n/2 budget or 2n budget.
* "we [ ] empirically prove", "provide experimental proof", "this proves claim 2": one cannot prove something empirically. We can empirically disprove a hypothesis, or provide some evidence supporting a hypothesis. But proving a hypothesis is impossible empirically. Please, rephrase.

Minor Qs/comments:
* I'm curious, why did you choose to work on DNCs instead of other alternatives (e.g. GNNs or transformers)?
* Interestingly, for Figure 1b, the non-adaptive curve is higher than the adaptive one for n_edges>160. What do you think happened there?
* Could you explain what 'model learns to perform the algorithm rather than describe it' mean?
* nit: Figure 1 is easy to parse when reading from a black&white print out, Figures 2, 4 and 5 are impossible to parse. Maybe increasing the space between the dashes or adding other line markers might help?


## Post rebuttal update
I appreciate the authors addressing some of my claims and raise the score from 3 to 5. However, I still think the paper requires more work to be published at ICLR: story coherence, DNC exposition, new experiments seem to affect the 'adaptive computation -> generalisation' claim.

---

> ### Author Response · Authors · 2023-11-22
>
> First, we wish to thank you for your review of our paper.
>
> 1. Story incoherency: we think the main driver of our paper is the intuitions gleaned from classical algorithms and complexity theory. This led us to the comparisons we did with other DNC works that only use a small constant number of planning steps, so we focused on that. Reframing our paper as focused on generalization would detract from our main takeaway, in our opinion. There are multiple other methods we could have looked at for better generalization. Here we’re specifically looking at the planning budget due to our complexity theory motivation, and observe generalization improvement due to that choice. We have also updated Section 3.1 to better explain our motivation.
> 2. We agree that it’s difficult to faithfully present the DNC in such a short paragraph as we have. We took inspiration from the way other papers in the field presented the DNC. An illustration of the DNC isn’t simple, and we felt that by adding one we’ll have to spend much more time explaining it than we’d like. Following your suggestions, as we wanted to incorporate the new experiments into the revised paper, we had to move this introduction to an appendix. This will allow us to elaborate further on the DNC model in the final submission if needed.
> 3. Loose phrasing:  Given the additional and improved evidence we supply in our revised version, we hope it gives better support to our usage of general language. \
> Importantly, we also improved our terminology, specifically around the notion of “more general usage of memory” - please see Section 3.3.
>     1. This is a very good point. The original DNC paper as well as following research used 10 planning steps or none at all, which is why we chose this value. To the best of our knowledge, no other DNC paper tried to increase this value. Our motivation was that any constant budget inherently poses a limit to the DNC model. We added Section 5.2, where we further discuss this as well as present results of new experiments with p=75 and p=200.
>     2. We agree that this statement is merely an intuitive explanation and not empirically demonstrated. For the purpose of our paper it doesn’t really matter what algorithm was learned, we simply gave a potential explanation as an example for one possibility. An experiment demonstrating it could be very interesting and we consider doing it for future work.
>     3. We agree here too. We still didn’t find the best terminology for what we want to say, however we feel that the current terminology is better as we explained in point 1 above.
>
> Answering your questions:
> 1. As we stated above, we’ve added experiments with p=75 and p=200, to improve our comparison between constant and adaptive budgets.
> 2. Both claims are explanations for different mechanisms of the DNC. The impact of both is the same - generalization, as you’ve pointed out. However the details of the mechanism are important, as they allow us to better explain the cause for the improvement. It allows us, for example, to increase the memory size during inference and expect an improvement. We think it’s not trivial that more planning steps also improve memory utilization, hence our separation of these two claims.
> 3. A great suggestion, unfortunately these runs have not concluded in the past week. However, for example, the empirically determined planning budget of the quadratic model, does not match the given planning budget (Appendix D). Furthermore, some of the larger constant runs also have an optimal budget different from the one provided in training (See Appendix F). These provide evidence that it is possible for the models to learn a solution requiring a different time than trained with.
> 4. You’re correct. We rephrased.
> 5. DNCs are more explicitly about algorithms than the other methods. We didn’t choose GNNs since we didn’t limit our results to graphs, and using transformers for algorithmic reasoning is a very interesting emerging field that we’re probably going to work on next.
> 6. We’re not sure. It’s very low performance anyway, so arguably could be irrelevant. The differences between the models aren’t big to begin with.
> 7. What we meant by this is to explain that DNC belongs to the implicit learning approach, where given a problem, the model doesn’t output an algorithm, but rather performs, or “simulates” the algorithm on the input in order to provide the result. Because of this, it’s hard to know what exact algorithm the DNC learnt.
> 8. We couldn’t find a good choice of dashes that looked good, so we chose to keep the color-centric presentation.

---

### Official Review · Reviewer_oFbH · 2023-10-31

**Soundness:** 3 good
**Presentation:** 3 good
**Contribution:** 2 fair
**Rating:** 5
**Confidence:** 4

**Summary:**

The authors hypothesize that Differentiable Neural Computers (DNCs) require adaptive planning budgets in order to generalize effectively to different input sizes. They claim that current methods that train DNCs with fixed p cannot learn more general purpose algorithms since the minimization of the training error often makes these existing approaches learn heuristics/non-general algorithms that cannot effectively generalize to OOD data. Moreover, this also means that such approaches cannot utilize additional memory effectively further limiting their generalization capabilities.

To show that adaptive planning strategies are necessary, the authors conduct an empirical evaluation on two graph problems: Shortest Path and MinCut. The authors propose a linear planning budget on the size of the input graph and show that DNCs training with such a planning budget are able to better generalize to graphs with larger sizes than seen during training. The authors also show that DNCs cannot learn generic algorithms if trained with a fixed budget and allowed a variable budget during inference.

Finally, the authors also show that such adaptive models can also better utilize addtional memory whenever it is available and thus generalize to larger inputs more effectively.

**Strengths:**

1. The paper is quite clear and provides adequate background on DNCs. The empirical section is well-explained and the organization of the paper is smooth.

2. The ideas put forward by the authors are intuitive. The runtime (and memory) complexity of algorithms is often a factor of the input and thus it is not surprising that neural networks that can mimic Turing machines might require something similar.

**Weaknesses:**

I think that the ideas are very interesting but I am afraid that the empirical evaluation might require more experiments for the authors to make the claim that adaptive planning budgets are needed. I post my questions here itself.

1) I appreciate the clarity of the plots in the paper but I feel that the total number of domains is rather limited. The authors run their experiments on only two problems and the performance differential is significant on only one of the problems (Shortest Path).

Q: Could you please motivate the choice of using a single performant domain for the analysis?

2) There is no analysis about the resources expended for training with the adaptive vs. fixed scheme. Simple plots or data showing that training times vs performance tradeoff would have enhanced the clarity of the paper.

Q: Do you have any data pertaining to the training time required for the adaptive case?

3) Building upon my previous statement, currently there is no way to automatically determine the adaptive planning budget but relying on some human-expert input such as "the total number of edges".

Q: Do you have any general intuition as to how to better select the adaptive planning budget as opposed to an empirical approach? (I recognize that this is a hard problem in itself but I would like to know if an easier training scheme exists)

4) For Fig 3b: The purple (edges=40) line drops starkly in performance when given more planning steps. ie. the adaptive scheme was not able to maintain steady state performance (as the authors put it in Sec 4.2)

Q: Do you have any intuition as to why this happened for the purple case?

**Questions:**

I've listed my questions to the authors under the section on Weaknesses. I hope that the authors can help clarify my concerns.

---

> ### Author Response · Authors · 2023-11-22
>
> First, we wish to thank you for your review of our paper. \
> We now answer the questions presented:
>
>
>
> 1. We agree that showing experiments for more problem domains is important. Unfortunately, compute resources and runtimes were the main limiting factors in this work. Especially for a more complex problem like MinCut, training with large planning budgets becomes prohibitive very quickly. A lot of the time was spent on optimizing the training code to reduce this overhead, but we still couldn’t completely remove this problem.  \
> We have updated Section 3.1 to better explain our motives in choosing these specific problems. We believe most problems previously used to evaluate DNCs are too simple and therefore not relevant, as they don’t fall in the category of algorithms that would benefit from an adaptive planning budget.  \
> We also wish to state that most DNC papers do a rather “shallow” exploration of the problems they evaluate on. We tried to do a deeper dive into the mechanism of the model as exhibited in our chosen problems. For example, choosing the correct data distribution to prevent the model from overfitting was a laborious task. This prevented us from examining more problems in time for this submission.
> 2. This is a very good question. We ran some further experiments including suggestions from other reviewers, which address this question directly. It is now explained in Section 5.2, including a figure comparing the generalization performance of models with different training budgets vs. the compute used during training. Note that “compute” here is a bit nuanced, please refer to the updated section for details. Specifically, Figure 6(b) shows that training resources used to train the adaptive model match the constant baseline of p=10. Furthermore, to reach the same performance as the adaptive model with a large constant budget, it required quadruple the amount of computation resources during training.
> 3. This is indeed a very hard problem to solve generally. We can say two things here: first, future work could tackle this question directly; second, if we estimate the complexity of a problem we could use the asymptotic complexity as the function for the size of the planning budget. For example, the Shortest Path problem is known to have linear complexity, hence our choice for it makes sense. For MinCut, the best known algorithm is |V|  |E|, which motivates us to use that value. We will be adding this run to the final submission.
> 4. We don’t have a good intuition for this effect yet. This question is interesting, and points at potential nuance in the behavior of the DNC model. Answering it might lead to further improvements. However, note that the rate of degradation is approximately linear with the number of extra planning steps, in contrast with the almost instantaneous reduction that occurs for the constant model. Even though the adaptive model’s steady-state is not as stable for small inputs as it is for large inputs, it’s still much more stable than the constant model.

---

> > ### Comment · Reviewer_oFbH · 2023-11-23
> >
> > Thank you for your responses.
> >
> > I agree that finding an automatic way to find the best planning budget is orthogonal and interesting work.
> >
> > I do agree that computational resources can be limited especially considering the domains run but I feel that the work would need to showcase significant gains on at least more than 1 domain as is currently there in the paper. The claim that other problems in current literature are too easy and would not benefit are strong points for your paper but then it does mean that the onus is on you to show that adaptive planning does not worsen performance. Extrapolating and stating conclusions from just one domain might not be sufficient.
> >
> > Overall, I am happy with the paper and I think that it is interesting. I've decided to keep my current score since I feel that a single domain is not sufficient but I will have no problem if this paper were to get accepted. It solves an interesting problem in a novel fashion.

---

### Official Review · Reviewer_vfng · 2023-11-01

**Soundness:** 2 fair
**Presentation:** 3 good
**Contribution:** 2 fair
**Rating:** 3
**Confidence:** 3

**Summary:**

The paper leverages differentiable neural computers (DNCs), and proposes to improve their generalization capabilities by training them with a flexible number of planning steps, i.e. computation steps between processing the inputs and producing the outputs. Experiments on shortest path and min-cut graph reasoning problems show that a DNC trained with a number of planning steps that is linear in the input size instead of constant, can generalize better to graphs that are larger than those seen during training and leverage differentiable memory modules that are larger than those it was trained with.

**Strengths:**

The paper is well-written and easy to follow. It is interesting to see that the DNC models learn algorithms that can indeed generalize beyond the graph sizes seen during training (both with and without adaptive planning time). The introduced scheme to allow DNCs to leverage larger memory modules at training time by adding a temperature rescaling to the softmax over memory slots is intuitive and a nice auxiliary contribution.

**Weaknesses:**

The main claim of the paper is that the DNC can only learn a generalizable algorithm because it is trained with a flexible planning budget. In other words, the paper claims that one cannot learn a generalizable algorithm with a DNC trained with fixed planning budget. To prove this, the paper compares a DNC trained with a fixed planning budget of 10 steps, to a DNC trained with a flexible budget, equivalent to the size of the input. In the experiments, both DNCs are trained on input graphs with up to 75 edges.

I have two problems with this argumentation:

1) the DNC with constant planning budget of 10 steps has much less computation available than the DNC with flexible planning budget, which uses up to 75 steps during training. At test time, the difference becomes even larger as we evaluate the DNCs on graphs with up to 200 edges. It does not seem fair to compare models with such different compute budgets.

2) It may be that the fixed planning budget of 10 steps is simply too small to learn general algorithms. To prove the point that flexible planning budgets are required for learning generalizable algorithms, the authors should show that a baseline with a fixed planning budget **equivalent to the longest graphs** the model is evaluated on (i.e. 200 steps in the experiments) still fails to learn generalizable algorithms.

For the latter point, one could argue that it is impractical or wasteful to train with the max planning length even for small inputs. But still, the fixed planning budget should be set in a way that it matches the compute flops used for the flexible planning model during training. It is unclear how the current value of 10 was chosen and it seems to result in a baseline that has overall less compute and thus an unfair comparison.

The same argument holds true for the "increased memory utilization" experiments.

A separate concern is the stability of the proposed approach. The results in appendix C suggest that the models' generalization capability heavily varies between different seeds. This is concerning since the model presented in section C does not exhibit the central generalization capabilities that the paper claims.

Finally, the paper uses the simple heuristic of setting the number of planning steps to be equivalent to the length of the input. This is an arbitrary choice that may not generalize well to other tasks and may require hand-tuning on any new task family.

**Questions:**

- It would be interesting to understand whether similar benefits of adaptive computation / planning time can be observed in modern transformer models, e.g. by giving LLMs the ability to ponder for a number of steps that's proportional to some measure of the input complexity before producing output tokens.

- How does the training compute used for the DNC with fixed planning steps compare to that of the DNC with adaptive planning steps?


=========================
# Post-Rebuttal Comments

Thank you for answering my review.
It seems that the authors agree with the points in my review -- the new experiments with longer *fixed* planning length seem to change the main claim of the paper from "adaptive planning budgets are required to learn generalizable algorithms" to "adaptive planning budgets make training more compute efficient", which is a reasonable claim, but maybe less surprising. I also skimmed the other reviews and all of them seem to agree that the current paper does not meet the bar for acceptance. Thus I maintain my score.

**Details Of Ethics Concerns:**

--

---

> ### Author Response · Authors · 2023-11-22
>
> First, we wish to thank you for your review of our paper.
>
>
>
> 1. First, we agree with your observation that comparing our adaptive method with a constant planning budget of size p=10 may be unfair due to the difference in computation available to the models. All current work on DNC, to the best of our knowledge, uses a constant planning budget of p=10 or none at all, which is the justification for that choice for this specific constant for the baseline. This is a detail which we have accidentally removed from the first version of the paper. We made sure to clearly state this in our revised version, in multiple locations.
> 2. Throughout the paper we describe how this choice hinders performance significantly. So, we ran a few experiments in the past week, where we trained two models with larger constant planning steps, one with p=75,which is equal to the size of the largest input seen during training, and another with p=200, the size of the largest input seen during evaluation.
> 3. The results from this experiment affirm that a larger constant planning budget helps the model’s performance, which aligns with our basic premise, which is also the title of the paper - DNCs require more planning steps. However, the results also point out the natural limitations of such models. To make this clearer, we modified the explanation in Section 3.1, and we analyzed the results and compared them to the adaptive budget in the newly added Section 5.2. Notably, the results also demonstrate that the resources required for training the adaptive linear model match those required for the baseline p=10 model. Moreover, to match the performance of the linear model a constant of at least p=200 is required, which comes at the cost of quadrupling the training resources.
> 4. As for the memory utilization – the reviewer had a concern that a constant model with a bigger memory is missing, but it is not feasible, resource-wise.
> 5. Regarding the instability with respect to the random seed as presented in Appendix C, we also observe this phenomenon with some of the constant budgets. Hence this property does not affect our comparison, it is simply a property of training a DNC.
> 6. Regarding the heuristic choice of adaptive budget, we agree that searching for an optimal adaptive budget scheme (rather than linear) might be required per task, and could be an interesting direction for future work. We will be adding an adaptive p=|V||E| model to the MinCut in the final submission to showcase other adaptive schemes.
>
> Regarding your questions:
>
>
>
> 1. No doubt, giving the same tools to transformers is a direction to think about, both in “planning time” and “memory utilization”. By choosing DNCs we are able to directly evaluate and reason about their usage and utilization of memory, which is crucial when discussing them as a method to find general algorithms. Transformers on the other hand do this implicitly within their latent spaces, making this analysis much more complex.
> 2. Regarding the training compute, we discuss this in Section 5.2 which we added, specifically Figure 6(b) shows the training resources used to train the linear model match those required to train the p=10 baseline.

---

### Official Review · Reviewer_oJyN · 2023-11-04

**Soundness:** 3 good
**Presentation:** 3 good
**Contribution:** 2 fair
**Rating:** 3
**Confidence:** 3

**Summary:**

The paper proposes an extension to the differential neural computer (DNC, originally presented in a 2016 Nature paper) which proposes an end-to-end differentiable architecture based on LSTMs for learning algorithms from input/output example. Focusing on one of the original experiments (the classical graph problems shortest path and min-cut), the idea presented in this submission is to introduce variable processing time (LSTM steps) between inputs and outputs (for these problems, the original DNC used a fixed amount of 10 processing steps. This processing time is computed based on manually specified dependency on the input length. The authors perform experiments with a setting the processing time to the input length, which works well for the shortest path problem (as in, the trained model demonstrates better extrapolation to problem sizes not seen during training compared to a fixed number of processing steps. In min-cut, the improvements are less significant.

**Strengths:**

The paper is, for the most part, clearly written and presents the idea in an easy-to-understand manner. The authors devote a lot of room for additional experiments that aim to investigate how adaptive processing time improves extrapolation. They also present interesting results on how extrapolation to a large amount of memory can benefit from an adaptive processing time which appears to have been a long-standing issue in DNC extensions.

**Weaknesses:**

The contributions of the paper concern shortcomings of the DNC, which, at least to my knowledge, has not gained a strong foothold in learning general algorithms from data. While it's refreshing to see non-LLM submissions, I'm afraid that the paper at hand might not be of high interest to the community at the moment.

The paper is built around two claims (adaptive planning budget during training leads to learning more general algorithms, and adaptive planning budget allows for memory usage that generalizes better) that are marked as important and general insights, whereas they require 1-2 paragraphs regarding the specific settings they apply to. I would like it better if the formatting would be tuned down a bit given the experimental evidence.

For claim 1, I do not see how the paper supports clearly demonstrates how an adaptive planning budget generalizes to larger inputs in the general case. While the idea in itself makes sense and does work better on shortest path, there's no significant benefit on min-cut (the authors give a good explanation why that's the case, but in the end there's no evidence for the claim on this problem). More importantly, there is no baseline taking, let's say, a fixed amount of 100 planning steps instead of 10. The claim also seems to leave out the part where, with an adaptive planning budget, the model appears to perform worse on short inputs. Lastly, the claim focuses on using adaptive processing time during training, whereas it seems to required during testing as well (and that's not mentioned in the claim).

For claim 2, I found the formulation a bit ambiguous. What do you mean with "use its memory addresses in a more general way"? I also don't think there's enough evidence to support this claim, assuming it means that the resulting model can benefit from increasing the memory size at inference time. In contrast to what the authors write in the last paragraph of section 5, the memory reweighting scheme  also benefits the DNC with a fixed amount of processing steps (Fig 5a); this should become clear if the figure is being zoomed in. To me it seems like we see similar improvements, relatively speaking, for fixed and adaptive planning budgets.

In summary, my view is that (1) the paper concerns a topic of marginal interest; and (2) the claims as they are made to be are not sufficiently supported by the evidence that's presented.

**Questions:**

I was wondering why the fact that the DNC is already taking an adaptive number of steps based on the problem size is not discussed? What is the sensitivity of the DNC to the amount of processing steps in general? Is 10 just a good number for the problem sizes considered in the original paper?

Regarding the requirement for quadratic processing time on min-cut: have you considered training and testing on smaller instances of the problem instead?

Please fix the citations so that braces are used when they occur in text but are not part of a sentence.

---

> ### Author Response · Authors · 2023-11-22
>
> First, we wish to thank you for your review of our paper.
>
>
>
> 1. Regarding the settings in which our claims apply: the settings in which our intuitions apply are very broad. Our explanation was in the spirit of complexity theory papers, which we agree can be made more clear. We modified the explanation we presented in Section 3.1 to include the distinction between problems that can be solved using on-line algorithms with constant latency and those that cannot, rather than discuss bounds. Hopefully it’s clearer now.
> 2. For Claim 1:
>     1. First, we accidentally omitted the origin of the value 10 for the baseline. It’s the value presented in the original paper of the DNC, as well as the value used in later DNC related works. To our knowledge, no other work tried to increase this value, and this is one of the main motivations for our work. We’ve added this clearly to the paper, in several locations.
>     2. We agree with your observation that comparing with constant p=10 is problematic. This helps emphasize the basic premise, which is also the title of our paper: DNCs require more planning steps. A better comparison would be first between two constant models, one with the value 10 from literature and another with some higher value. Then we can compare the model with a large planning budget against a model with an adaptive budget. So, we ran a few experiments in the past week, namely two models with constant planning steps, one with p=75 and another p=200. We’ve updated our paper to include these new experiments in Section 5.2.
>     3. You’re correct, our method assumes the same budget both during training and during inference. We ask the question “can we use a different budget during inference” in Section 4, though not in so many words. We added a clarification to the paragraph, stating that the results in this section indicate we need to use the same training budget for inference as well.
> 3. For Claim 2: \
> Ambiguous formulation: We reformulated our claim to reduce ambiguity. \
> In our updated definition, a “more general use of memory” is split into three different aspects of memory utilization:
>     1. How long can a model hold onto memory? (long-range dependencies)
>     2. How effectively can the model manage a larger memory during inference?
>     3. How efficient is its usage of memory with respect to input size?
>
>     These aspects can be measured. We modified Section 4 and Section 5 to support this claim more clearly, along with additional data to support the last aspect, added to the new Section 5.2. As a part of this modification, we cleared up the last paragraph in Section 5 (now 5.1). The constant model’s generalization doesn’t improve at all when we don’t reweight the memory access distribution, which is what we meant to claim.
>
>
> We now directly answer the posed questions:
>
>
>
> 1. We aren’t sure we understand this question, so we answered it in two ways.
>     1. If you mean that existing work already uses an adaptive planning budget: we aren’t aware of such work. To the best of our knowledge, all work on the DNC model uses a constant number of planning steps or none at all. The constant value is 10 in all papers we found.
>     2. If you’re referring to the length input phase which is indeed adaptive, then this is now better described as per our discussion of on-line algorithms with constant latency in Section 3.1.
> 2. It is very difficult to answer generally, rather should be answered on a per-problem basis. In Section 4 we show that a DNC trained with a small constant planning budget is very sensitive to the amount of planning steps, while the DNC trained with a linear budget has a phase transition; for a small range of planning steps, it has high sensitivity, and everywhere else the sensitivity is low (before or after the phase transition).
> 3. No, it is the number considered in the original paper. It isn’t good, as demonstrated by our results for Shortest Path. The problem sizes we used for that problem are identical to the original paper, and we significantly improve generalization as shown by our results. We’ve also added an analysis in the newly added Section 5.2 that it is equivalent in compute time to the adaptive model during training, while to match its performance with a constant budget we need about X4 compute (when p=200).
> 4. Yes, we have. Unfortunately, for very small graphs the min-cut problem becomes uninteresting. The variety of different cuts is very reduced, and the simple rule of “remove the node with lowest degree” is usually the correct cut. Only in larger graphs is this rule no longer a good choice. When we trained models on smaller problems we saw this exact behavior, and were forced to increase the size of training graphs.

---

### Meta-Review · Area_Chair_NSCj · 2023-12-09

**Metareview:**

This paper proposes an extension to the well-known Differentiable Neural Computer paper (which learns a memory-augmented neural network architecture to imitate general algorithms), by introducing an adaptive number of planning steps that scales with input size during training. The paper compares a fixed number of planning steps on a number of graph shortest-paths and mincut problems. For shortest paths the adaptive planning steps make significant difference and show better generalization, but for mincut, they do not. Reviewers were hoping to see results on a larger family of algorithms than these two. The paper makes a secondary claim that the use of the adaptive planning steps leads to more efficient memory use that scales roughly linearly with the input size (according to Fig 7), while the fixed planning steps use a large amount of memory, regardless of the memory size. This secondary property of the adaptive planning steps is interesting, but again would benefit from more experimentation on additional algorithms. Without solid evaluation of the generalization properties of the proposed method, the secondary property of memory efficiency is not sufficient in my opinion for the paper to stand on its own. So, I would advise the authors to do one more iteration of this work and focus on the primary claim.

**Justification For Why Not Higher Score:**

This paper is interesting, but needs further experimental validation.

**Justification For Why Not Lower Score:**

N/A

---

### Decision · Program_Chairs · 2024-01-16

Reject